

# Introducing Ice Nucleating Particles functionality into the Unified Model and its impact on the Southern Ocean short-wave radiation biases

Vidya Varma[1], Olaf Morgenstern[1], Kalli Furtado[2], Paul Field[2], and Jonny Williams[1]

[1]National Institute of Water and Atmospheric Research (NIWA), Wellington, New Zealand
[2]Met Office, Exeter, United Kingdom

**Correspondence:** Vidya Varma (vidya.varma@niwa.co.nz)

**Abstract.** Insufficient reflection of short-wave radiation especially over the Southern Ocean region is still a leading issue in many present-day global climate models. One of the potential reasons for this observed bias is an inadequate representation of clouds. In a previous study, we modified the cloud micro-physics scheme in the Unified Model and showed that choosing a more realistic value for the capacitance or shape parameter of atmospheric ice-crystals, in better agreement with theory and observations, benefits the simulation of short-wave radiation over the Southern Ocean by brightening the clouds. However, attempts to modify the cloud phase by directly adjusting the micro-physics process rates like capacitance tend to affect both the hemispheres symmetrically whereas we seek to brighten only the high-latitude Southern Hemisphere clouds. In this study we implement a simple prognostic parametrisation whereby the heterogeneous ice nucleation temperature is made to vary three-dimensionally as a function of the mineral dust distribution in the model. As a result, those regions with less dust number density would have lower nucleation temperature compared to the default global value of $-10°C$. By using mineral dust as an indicator for ice nucleating particles in the model, this parametrisation thus captures the impact of ice nucleating particles on the cloud distribution due to its general paucity over the Southern Ocean region. This approach thus improves the physics of the model with minimal complexity.

## 1 Introduction

The long-standing issue of insufficient reflected short-wave (SW) radiation over the Southern Ocean (SO) region in climate models due to an inadequate representation of clouds has been identified as a significant concern in the 5[th] Assessment Report of the Intergovernmental Panel on Climate Change (IPCC AR5; Chapter 9) (Flato et al., 2014). Previous studies have shown that cloud representation in the model can be improved through modifications to mid and low-level clouds (Bodas-Salcedo et al., 2012, 2014), the shallow convection scheme (Kay et al., 2016), mixed-phase clouds (Furtado et al., 2016; Lohmann and Neubauer, 2018) and atmospheric ice micro-physics parametrisation (Furtado and Field, 2017; Varma et al., 2020), which all reduce the radiation biases.

A major factor that influences the brightness of clouds in the models is the supercooled liquid water (SCL) content (Bodas-Salcedo et al., 2016; Furtado et al., 2016; Engdahl et al., 2020). By targeting the sinks of SCL, its presence in the model





can be controlled thus improving the cloud cover. In an earlier study, we noted that choosing a more realistic value for the
capacitance or shape parameter of atmospheric ice-crystals, in better agreement with theory and observation, has an impact in
reducing the SW radiation biases over SO region. Using a shape parameter for ice crystals such that non-spherical particles
are also considered, results in a decrease in the depositional ice growth reducing the Ice Water Path (IWP) (Fig. 1a). This is
accompanied by an increase in the liquid water path (LWP) over the SO region (Fig. 1b) which benefits the simulation of SW
radiation (Fig. 1c) by brightening the clouds due to more liquid water content (Varma et al., 2020). The capacitance essentially
acts as a sink for the SCL and reducing its value slows down the depositional growth of ice particles from water vapor, which
leaves more water vapor to be available for condensation into liquid phase particles. However, attempts to modify the cloud
phase by directly adjusting the cloud micro-physics process rates (like capacitance) tend to affect both hemispheres uniformly,
whereas observations indicate that only the Southern Hemisphere (SH) high-latitude clouds need to be brightened. Similar to
capacitance, another potential domain that can act as a sink for SCL is that of the various ice building processes in the model.

Ice forms in the atmosphere through homogeneous and heterogeneous ice nucleation processes (e.g. Pruppacher and Klett
(2012)). Ice nucleation in the troposphere due to homogeneous freezing can occur at temperatures as low as $-40°C$, without
any Ice Nucleating Particles (INPs). INPs are those atmospheric aerosol particles which initiates and facilitates the freezing
of water in clouds. However, in the presence of INPs, ice formation can proceed quicker through heterogeneous nucleation
(Pruppacher and Klett, 2012; Kanji et al., 2017). The various heterogeneous ice nucleation processes in the atmosphere are
that of deposition nucleation (when water vapor deposits directly on a deposition nucleus), immersion freezing (when an ice
nucleus is present within the drop), contact freezing (when an ice nucleus makes external contact with a supercooled drop thus
quickly initiating freezing) and condensation freezing (when a transient water drop is formed before the freezing occurs, and
then freezing occurs via either contact or immersion nucleation) (Curry and Webster, 1999; Pruppacher and Klett, 2012; Kanji
et al., 2017). Our focus will be on the immersion freezing process as it is the most commonly implemented heterogeneous ice
nucleation process in global climate models (GCMs).

Since in-cloud micro-physics and convective clouds are treated separately in almost all of the low-resolution GCMs, immersion freezing is also parametrized separately in micro-physics and convection schemes. While in-cloud micro-physics predicts
the cloud phase, the convection scheme provides a temperature dependent threshold for the detraining of ice (e.g. Kay et al.
(2016)). As a result, along with the micro-physics scheme, the convection scheme also plays a role in determining the ice
formation in the model through detrainment temperatures.

For the immersion freezing process to take place, the presence of an INP is required to act as the nucleus for the formation
of an ice crystal in the atmosphere. Kanji et al. (2017) gives a comprehensive overview of the primary sources and types
of atmospherically relevant INPs such as mineral dust, metals/metal oxides, bioaerosols, soil dust etc. Among these, mineral
dust is an ideal candidate for being the most common and effective INP and has been implicated in the generally low-INP
environment over the SO region. However, this INP dependency on immersion freezing is not included in most of the GCMs
but rather immersion freezing is modelled using a single temperature dependent freezing parametrisation (e.g. Fletcher (1962)
in our model).





In this study, we propose a simple parametrisation whereby the immersion freezing temperature in the model is linked to the prognostic mineral dust distribution through a diagnostic function, resulting in those regions with more (less) dust to have warmer (colder) nucleation temperatures, thus resulting in regional differences in the nucleation temperatures. This provides a functionality to mimic the role of INPs in the atmosphere on influencing the SW radiation over the SO region by impacting the cloud phase. Several recent studies have shown the significance of INPs in influencing the cloud radiative properties (DeMott et al., 2010; Vali et al., 2015; Kanji et al., 2017; McGraw et al., 2020). Vergara-Temprado et al. (2018) showed that in a regional high-resolution model with double moment configuration of the same model that we use in our study, reduction of INP concentrations can increase the LWP and SW reflectivity over the SO region. Hoose et al. (2010) implemented an ice nucleation parameterization based on classical nucleation theory, with aerosol-specific parameters derived from experiments, into a global atmospheric model and has shown that immersion freezing by mineral dust is a globally dominant ice formation process.

In order to thoroughly examine the impact of INPs on cloud radiation properties, ideally state-of-the-art atmospheric models with extensive double-moment bulk micro-physics schemes and comprehensive interactive aerosol chemistry are desired. However, for GCMs as well as weather prediction models, this would be more computationally expensive. To bypass these technical barriers and yet understand the impact of INPs on climate, we have adopted an approach whereby the diagnostic nucleation temperature is linked to the prognostic dust.

## 2 Model

The control climate model used in this study is the most recent version of the Met Office's Unified Model, GA7.1 (Walters et al., 2019) however with modified micro-physics scheme for riming process and several other scientific changes. Appendix A summarizes the scientific set-up for the model version used here. The resolution used here is N96L85 (i.e. a horizontal resolution of $1.875° \times 1.25°$ and 85 terrain-following hybrid-height levels extending to 85 km of altitude). It uses the "ENDGAME" dynamical core with a semi-implicit semi-Lagrangian formulation to solve the non-hydrostatic, fully compressible deep-atmosphere equations of motion (Wood et al., 2014). The control model also follows the Atmospheric Model Intercomparison Project (AMIP) experimental protocol (Gates et al., 1999), using prescribed sea surface temperatures. The control model that was used by Varma et al. (2020) is a scientifically identical version of the control model used in this study.

## 3 Prognostic-dust parametrisation : The physical link

The temperatures that determine ice formation in the model are the homogeneous ($thomo$) and heterogeneous (i.e. immersion freezing) ($thetr$) nucleation temperatures. The global default value in the control model when the heterogeneous nucleation of ice first starts to occur (i.e. $thetr$) is $-10°$C. This is however not a realistic value that can be used globally because of the paucity in INPs in clean environments such as SO. As a result, in reality, the onset of ice via heterogeneous nucleation is displaced to colder values of $thetr$ for these cleaner regions compared to other aerosol-abundant regions like the Northern





Hemisphere (NH). Since heterogeneous ice nucleation is solely following the temperature dependent function suggested by
Fletcher (1962), there is no INP dependency taken into account in the control model. A rational way to introduce a more
realistic and targeted distribution of $thetr$ is to make it dependent on mineral dust that has a strong hemispherical asymmetry
(Fig. 2). Generally speaking, the number density of dust is lower over the SO region by two orders of magnitude compared to
the NH (Fig. 2).

By linking $thetr$ to dust, ice nucleation is suppressed over the SO region. This results in an increase in the SCL amount
available over the SO region. Ideally, this would result in an increase in the in-cloud albedo due to more SCL clouds over
this region and thus yielding more outgoing SW radiation from top-of-the-atmosphere (TOA). However, there are additional
feedback mechanisms that could alter this path in the model (e.g. feedbacks from the convection scheme which are described
below).

As mentioned before, along with the nucleation temperature from the cloud micro-physics scheme of the model, the con-
vection scheme also plays a role in controlling ice formation in the model. The detrainment temperatures in the convection
scheme are those two temperature values at which a) detraining condensate as ice begins in the model, called $startice$ (the
default global value = $-10°C$) and b) all condensate is detrained as ice, called $allice$ (the default global value = $-20°C$).
These detrainment temperature values are also linked to the dust distribution with a linear ramp of the fraction of water mass
detrained as ice between these two temperatures. As a result, the modification to $thetr$ from micro-physics is also reflected
in the convection scheme and thus will have a collective and realistic impact on ice growth in the model. As a result, in the
prognostic-dust approach, both cloud micro-physics and convection schemes depend on the mineral dust distribution.

The prognostic-dust approach is demonstrated in the schematic representation (Fig. 3). Further technical details on the
implementation of this approach are given under Section 3.1.

### 3.1    Prognostic-dust parametrisation: Experimental design

As mentioned in Section 3, the mineral dust distribution in the model links to both the cloud micro-physics scheme (through
$thetr$) as well as the convection scheme (through the detrainment temperatures $startice$ and $allice$).

### 3.1.1    Stratiform cloud-microphysics scheme

In order to activate the diagnostic $thetr$ based on prognostic dust method, we designed an empirical equation by which those
regions with less dust density have colder nucleation temperatures. Accordingly, the value of $thetr$ is defined to vary three
dimensionally as an arc-tangent function of the mineral dust distribution in the model as,

$$thetr_n = thomo + (thetr - thomo) \times (\frac{arctan(5 \times log_{10}(\frac{dustnd}{refdust}))}{\pi} + 0.5) \tag{1}$$

where $thetr_n$ is the new heterogeneous nucleation temperature which now varies as function of dust , $thomo = -40°C$ (the
default homogeneous nucleation temperature in the model), $thetr = -10°C$ (the default heterogeneous nucleation temperature
in the model), $dustnd$ is the total number of dust particles per $m^3$) and $refdust$ is a reference value from the total dust number





density, a tuning parameter. This $refdust$ parameter can be chosen such that for the low dust concentrations characteristic of
the SO, $thetr_n$ is colder. We would like to note that eq.1 is a heuristic parametrisation for quite a complex process in reality. It
simply allows a smooth transition between low/high INP environment based on the dust distribution. Although, eq.1 is heuristic
at this stage, with adequate observations, the parameters could in principle be established. This step is foreseen in the future
developments.

The control model uses the CLASSIC dust parameterization which predicts mass mixing ratios for six particle size divisions
in the range 0.3–30 $\mu$m radius (Woodward, 2001). Using the representative diameters for each of the bins, the number density
for each bin ($dustnd_{bin}$) can be calculated using the Ideal Gas Law equation as,

$$dustnd_{bin} = dust_{mmr} \times (\frac{6}{\pi \times \rho_{dust} \times (drep)^3}) \times (\frac{P}{R_{spec} \times T}) \qquad (2)$$

where $dust_{mmr}$ = dust mass mixing ratio for each bin, $\rho_{dust}$ = density of dust particles (2650 kg/m$^3$), $drep$ = representative
diameter for each bin, $P$ = air pressure, $R_{spec}$ = specific gas constant for dry air (287.05 J/kg/K) and $T$ = air temperature. The
total dust number density is simply the sum of number densities of all six bins.

As a result of eq.1, $thetr_n$ now be varies between $-10°$C and $-40°$C (Figure 4).

### 3.1.2   Convection scheme

$thetr_n$ is passed to the convection scheme as phase fraction ramp temperatures for the detrainment. The convection scheme
consists of separate formulations for deep, shallow, mid and congestus convection. In the experimental set-up described in
this study, $thetr_n$ is passed as a three dimensionally varying array only through the shallow and deep level convections to
replace the detrainment temperatures. The mid-level and congestus convection schemes use the default ice nucleation values of
$-10°$C. The idea behind such an approach is to have a scheme which induces a minimal increase in complexity over the default
fixed $thetr$. We did conduct further experiments by passing three dimensionally varying $thetr_n$ values in mid and congestus
schemes as well (not shown). However, the maximum response is for a combination of shallow and deep levels and hence
discussing the results only from this set-up.

Finally the detrainment temperatures are replaced as $startice = thetr_n$ and $allice = (thetr_n$ - 10.0)$°$C so as to maintain the
original bridging value between the detrainment temperatures.

All simulations were run for 20 years (i.e. 1988 - 2008) under steady-state present-day conditions. The control model is
represented as $control$ and the prognostic-dust experiment model as $exp_{dust}$. The values used in the simulations are given in
Table 1.





## 4  Results

In this section, we follow the structure of the schematic (Fig. 3) in presenting the results.

After implementation of the parametrisation for INP functionality, owing to lower dust concentrations, $exp_{dust}$ now produces
colder nucleation temperatures over the SO region compared to the global value of $-10°C$ in $control$. This is shown in Figures 5a and 5b represented by the spatial and zonal distributions of $thetr_n$ as function of dust aerosol in the model. Figures 5c and 5d show the anomalies in the ice and liquid cloud condensates incremented via the convection scheme respectively. These values are a typical representation of the ice and liquid detrainment rates coming from convection scheme alone. As expected, the ice condensate shows a decrease (Fig. 5c) and liquid condensate shows a corresponding increase (Fig. 5d) in $exp_{dust}$ with
respect to $control$.

Figure 5e shows the anomaly in the heterogeneous nucleation rates from the micro-physics scheme between $exp_{dust}$ and $control$. As a result of colder $thetr_n$, the heterogeneous nucleation rates show a decrease over the SO region (Fig. 5e). The nucleation rate shown here is the rate of increase in the mass of ice due to heterogeneous freezing of liquid water droplets, resulting from the presence of INPs. Due to fewer INPs (or in this case, mineral dust as a surrogate for INPs), the heterogeneous
nucleation rates show a decrease over the SO region in $exp_{dust}$ with respect to $control$ as anticipated. This results in an increase in the SCL over the SO region (Fig. 5f). Also, in the tropics, an increased LWP is seen even though heterogeneous freezing has gone up, probably accounting for more liquid being detrained and frozen in the stratiform scheme than before (Figs. 5e and 5f).

Figures 5g and 5h represent the anomaly in the annual mean distributions of IWP and LWP respectively for the stratocumulus
boundary layer clouds in $exp_{dust}$ with respect to $control$. Compared to the earlier capacitance experiment ($exp_{cap}$) (Varma et al., 2020) and (Figs. 1a and 1b), the response is more confined to the SO region now (Figs. 5g and 5h), which is highly beneficial.

Resulting from the displacement of heterogeneous nucleation to colder temperatures, the ice growth over SO is now reduced (Fig. 5g) accompanied by the availability of more liquid water content (Fig. 5h) over the SO region. Due to the increase in the
LWP from increased availability of SCL, the in-cloud albedo generally shows an increase over the SO region as shown in Fig. 5i.

This could be expected to result in an increase in the outgoing SW radiation from TOA over the SO region. However, Fig. 6a shows an unexpected decrease in the outgoing SW from TOA in $exp_{dust}$ with respect to the $control$. This is due to the fact that there is now an overall decrease in cloud fraction (Fig. 6b). This is probably because, previously, the large amounts of ice
clouds were introducing compensating errors, which the new scheme now removes.

Further details on the potential feedbacks that could have contributed to the decrease in cloud cover over SO are discussed in Section 5.

As our primary focus is on the impact on SW radiation, detailed analyses of other radiative fluxes are not included in this study. However, the response of other surface and TOA radiative fluxes in $exp_{dust}$ has been included in the Supplementary
material with a brief discussion (Fig. S1).



## 5 Discussion

To further investigate the decrease in the total cloud cover instead of the anticipated increase over the SO region, in response to the prognostic-dust implementation, we compare the results from $exp_{dust}$ to that of $exp_{cap}$ (Fig. 1 and Varma et al. (2020)). In the comparison plots (Figs. 7 and 8), *control* remains the same but only experiments differ. Figure 7 represents the anomaly in the different cloud types between $exp_{cap}$ and *control* and Figure 8 represents the same but between $exp_{dust}$ and *control*. It could be noted that both experiments (i.e. $exp_{cap}$ and $exp_{dust}$) reduce the low cloud (e.g. Figs. 7g, 7h and Figs. 8g, 8h), but only in $exp_{cap}$ does this translate into a significantly thickening of the cloud layers and hence an increase in mid-level amount (Figs. 7d to 7f and Figs. 8d to 8f). Subsequently, the increase in the high-level clouds in $exp_{cap}$ (Figs. 7a to 7c) is not visibly robust in $exp_{dust}$ (Figs. 8a to 8c) especially in the top-thin and top thick clouds. Essentially, the increase in liquid cloud over the SO region that we achieved from prognostic-dust approach (e.g. Figs. 5d, 5f and 5h) is offset by the cloud fraction changes possibly arising from the impact of high/mid level clouds.

A possible factor is the potential feedback processes from the convection scheme in the model. As mentioned in Section 3, $thetr_n$ is calculated in cloud micro-physics scheme to replace the default value of $thetr$ and eventually passed on to the convection scheme as well. From our analyses (not shown), it was seen that response from the new parametrisation is robust when the detrainment temperatures are also replaced by $thetr_n$ in the convection scheme rather than just being used in the micro-physics scheme alone. Thus, a change in the impact could be achieved by modifying the cloud scheme through interactions with the convection scheme.

For instance, there are parameters in the convection scheme that could define the efficiency at which liquid and frozen cloud fractions for a given amount of detrained condensate is calculated. In the convection scheme, the detrained condensate mass is associated with a cloud fraction, and this fraction is generally diagnosed from the detrainment rates. However, this diagnosis is not treated symmetrically between ice and liquid clouds in the control model but rather assumes that ice-clouds are more spread out (or having a higher fraction) for the same condensed water content. In additional experiments ($exp_{eff}$ and $control_{eff}$), we have modified these parameters that define the cloud fractions for a given amount of detrained condensate such that they are diagnosed symmetrically in both $exp_{dust}$ and *control* . It shows that there is a slight increase in the mid-level clouds in $exp_{eff}$ (Fig. 9) with the added changes in convection scheme compared to that of $exp_{dust}$ (Fig. 8) and it thus also proves to have a more beneficial impact over the SO region in terms of outgoing SW from TOA. Figures 10a and 10b represent the anomaly for the austral summer (Dec-Jan-Feb) outgoing SW from TOA and total cloud cover from these new scenarios. It can be seen that both the outgoing SW at TOA and the total cloud cover show tendencies of increase over the SO region compared to Figs. 6a and 6b.

In the prognostic-dust approach implemented in our model, the SW distribution is impacted by the loss of cloud fraction due to changes in $thetr$ for the stratiform and convective cloud schemes. It appears that the liquid clouds are not covering as large an area as the ice clouds. This might be expected by considering the differences in the volumes of atmosphere above ice saturation compared to liquid saturation. Therefore, even though all of the chain of causal physical links described in Figure 3 have been satisfied, the final link to increasing the SW reflectance has been offset by cloud fraction changes. This final link

can potentially be improved in the future, but requires effort to both enable this link and yet not affect the performance of
the cloud across the rest of the globe. Most importantly, we have improved the representation of the physically constrained
part of the schematic. The missing links are due to arbitrary/unconstrained model aspects (e.g. the factors in the convection
scheme that could define the efficiency by which liquid and frozen cloud fractions for a given amount of detrained condensate
is calculated) so these should be improved or revised to accommodate the upgraded physics. Additionally this is one of the
first global atmospheric models to implement such an approach to simulate the roles of INPs with minimum complexity in the
micro-physics scheme to improve the SW radiation biases over the SO region. Further changes to the convection scheme to
address the feedback processes are part of our future model development.

## 6    Conclusions

A new parametrisation is implemented in the Unified Model whereby the heterogeneous ice nucleation temperature is made to
vary three-dimensionally as a function of mineral dust in the model. As a result, those regions with more (lesser) dust number
density would have warmer (colder) nucleation temperatures compared to the default global value of $-10°C$. This approach
provides a more realistic representation of nucleation temperatures in clean environments like SO region compared to the
more aerosol abundant regions. By linking the nucleation temperature to dust, the ice nucleation could be delayed over the SO
region resulting in an increase in the SCL amount available over the SO. However, the increase in liquid cloud achieved over
the SO region is offset by cloud fraction changes due to additional feedbacks arising from the convection scheme, impacting
the high/mid level clouds and finally resulting in losing some total cloud cover. Nevertheless, all the physical links are made
for the approach where we used dust as a prognostic for INP, except for the final fragment. This is mostly limited to our model
which could potentially be addressed through further changes to the convective parameters in the model that influence cloud
formation. To summarise, the prognostic-dust approach improves the physics of the model with minimal complexity.

*Data availability.*

Model data is available at: https://zenodo.org/record/4774578

## Appendix A

The control model used in this study is a scientifically identical version of the control model used in Varma et al. (2020).
Several changes were introduced in the control model used in this study relative to its predecessor GA7.1 (Brown et al., 2012;
Walters et al., 2019). These changes range from minor bug fixes and optimisations to major science changes in the convection,
large-scale-precipitation, boundary layer and radiation schemes. As far as our study is concerned, the main modification to
GA7.1 is the inclusion of the modified micro-physics scheme which includes a shape dependence of riming rates using the



parameterization by Heymsfield and Miloshevich (2003), as a measure to prevent small liquid droplets from riming (Furtado and Field, 2017).

Some of the modifications in the convection scheme include that of a prognostic based convective entrainment rate, implementation of a new melting scheme to remove larger spikes in convective heating in the mid-troposphere, a revised forced detrainment calculation and a corrected evaporation of convective precipitation to remove existing errors.

The modified boundary layer scheme also includes changes to reduce vertical resolution sensitivity and an improved turbulent kinetic energy diagnostic and how it is used for aerosol activation.

The change in the radiation scheme is the implementation of spectral dispersion suggested by Liu et al. (2008) to improve the simulation of the first aerosol indirect effect.

A brief overview of the science changes is available in the Unified Model Newsletter Dec 2017 edition (Research and Model Development News, pages 10 -12. This document is included along with the model data at: https://zenodo.org/record/4774578).

*Author contributions.* VV carried out the model runs, performed analysis, created figures, wrote the manuscript and was also involved in the design and conceptualization of the study. OM was involved with obtaining the project grant, supervised the study and analyses of results. KF and PF provided guidance and support in designing the model set-up and analyses of results. JW provided technical support in setting up global climate model in super-computer environment. All authors have read and approved the final paper.

*Competing interests.* The authors declare that they have no conflict of interest.

*Acknowledgements.* This work has been funded by the New Zealand Government Ministry for Business, Innovation, and Employment (MBIE) through the Deep South National Science Challenge. This work has also been supported by NIWA as part of its government-funded core research. The authors wish to acknowledge the use of New Zealand eScience Infrastructure (NeSI) high performance computing facilities, consulting support and training services as part of this research. New Zealand's national facilities are provided by NeSI and funded jointly by NeSI's collaborator institutions and through the Ministry of Business, Innovation & Employment's Research Infrastructure
programme. https://www.nesi.org.nz.





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

Atmospheric Chemistry and Physics Discussions — Open Access EGU

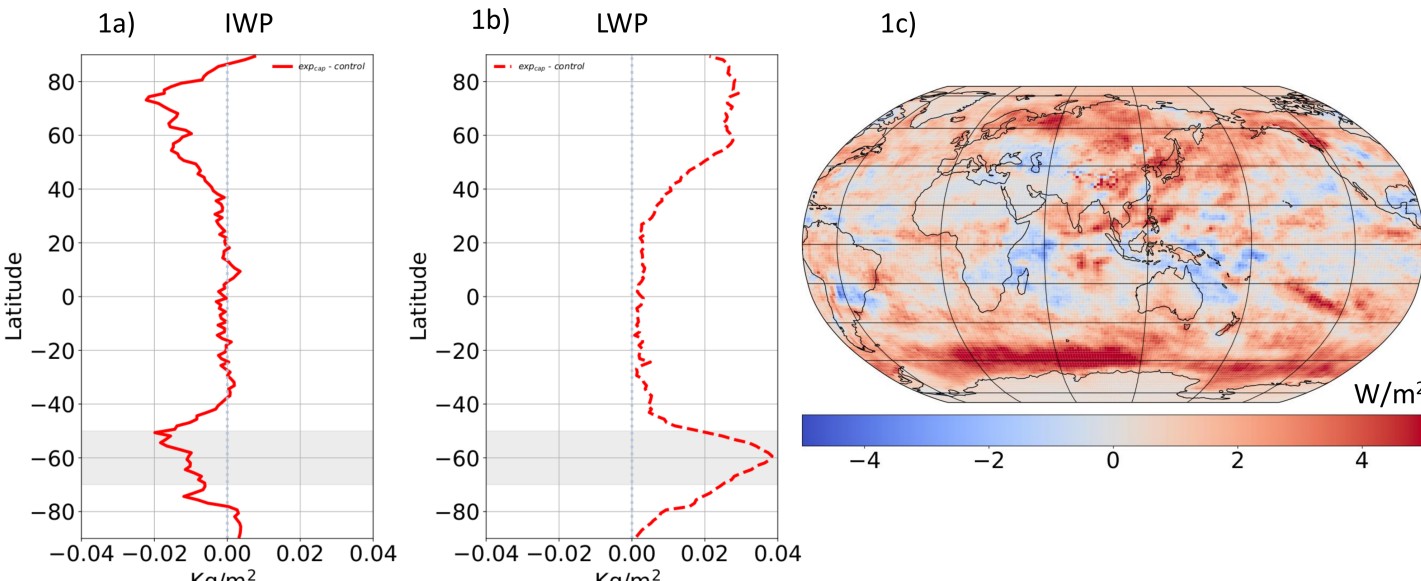

**Figure 1.** Annual-mean distributions of ice and liquid water contents along with outgoing SW at TOA in the capacitance experiment (henceforth $exp_{cap}$) from (Varma et al., 2020). (1a) Zonally averaged anomalies between $exp_{cap}$ and $control$ in the IWP over the stratocumulus boundary layer type clouds in the model. (1b) similar to (1a) but for LWP. There are seven boundary layer types that have been identified in the model based on the surface stability and capping cloud (Lock et al., 2000; Varma et al., 2020). As our focus is mostly on the stratocumulus boundary layer type clouds in this study, the cloud types considered in this figure are: type 2 = boundary layer with stratocumulus over a stable near-surface layer,type 3 = well-mixed boundary layer and type 4 = unstable boundary layer with a decoupled stratocumulus (DSC) layer not over cumulus. The IWP and LWP are calculated collectively over these types. Detailed description available in (Varma et al., 2020). (1c) Anomaly in the outgoing SW from TOA between $exp_{cap}$ and $control$. Values for (1a) and (1b) are calculated from 12 hourly instantaneous model output over 20 years and values for (1c) calculated from daily-mean model output. The SO region identified in this study is highlighted in gray in (1a) and (1b). All plots are produced from similar experimental set-up for both $exp_{cap}$ and $control$ mentioned in (Varma et al., 2020) but just using a more technically recent version of the Unified Model with similar science configuration.

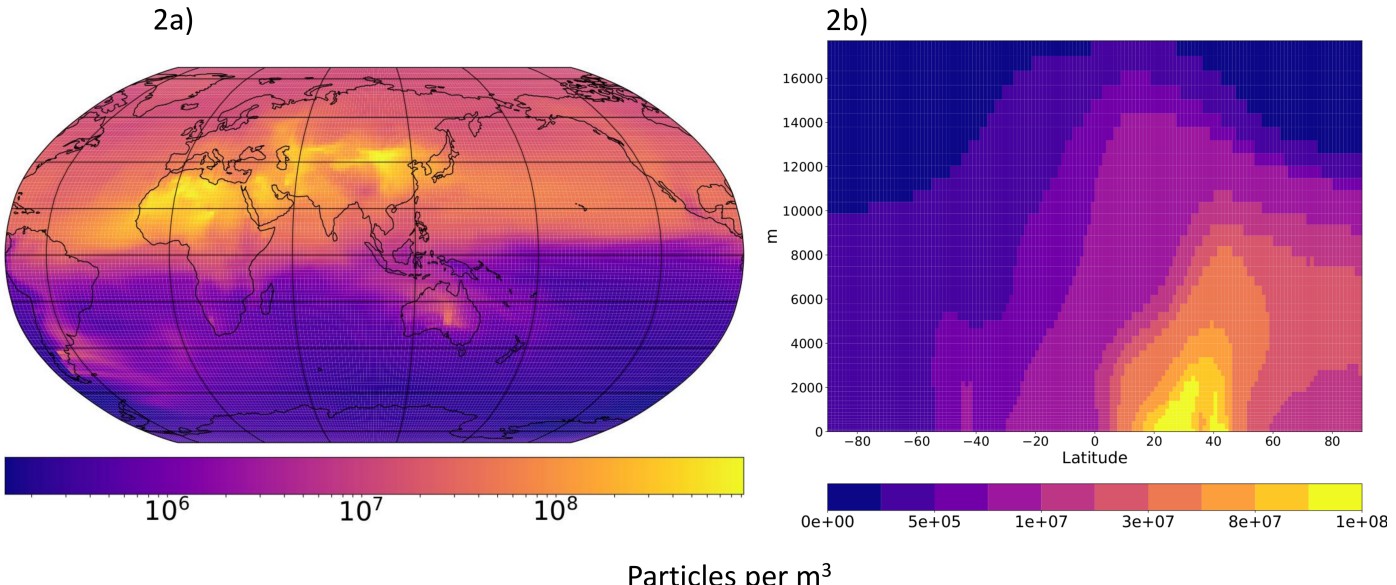

**Figure 2.** Annual-mean distribution of total dust number density in the control model. (2a) represents the spatial distribution of dust number density at the surface level (log scale). (2b) represents the zonal average depicting the vertical distribution. Values calculated from 10-day instantaneous model output over one year.



3)

**Less dust over SO region**

**Linking dust to ice growth determining temperatures in the model**

**Colder *thetr* over SO region**

**Onset of ice displaced to colder nucleation temperatures over SO region resulting in more available SCL**

*Optimal outcome*    *Additional feedbacks*

**Increase in total cloud cover**

**Decrease in total cloud cover**

**Increase in reflected SW radiation over SO region**

**Decrease in reflected SW radiation over SO region**

**Figure 3.** Schematic showing the physical links for prognostic-dust parametrisation



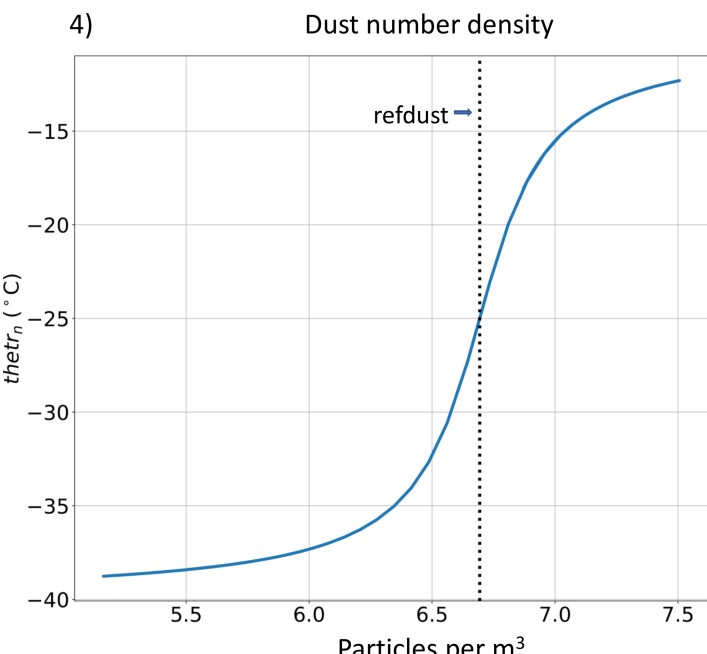

**Figure 4.** Representation of $thetr_n$ vs $dustnd$ (in $log_{10}$ scale) following eqn. (1). The dotted line represents $refdust$, an arbitrary reference value for annual-mean $dustnd$ which can be chosen such that the distribution of $thetr_n$ can be limited over preferred SH latitudes.


|         | $control$ (°C) | $exp_{dust}$ (°C) |
|---------|:--------------:|:-----------------:|
| $thetr$ | $-10$ | $thetr_n$ as $f(dust)$ |
| $startice$ | $-10$ | $thetr_n$ as $f(dust)$ |
| $allice$ | $-20$ | $thetr_n$ as $f(dust)$ - 10.0 |

**Table 1.** Values used in the model runs

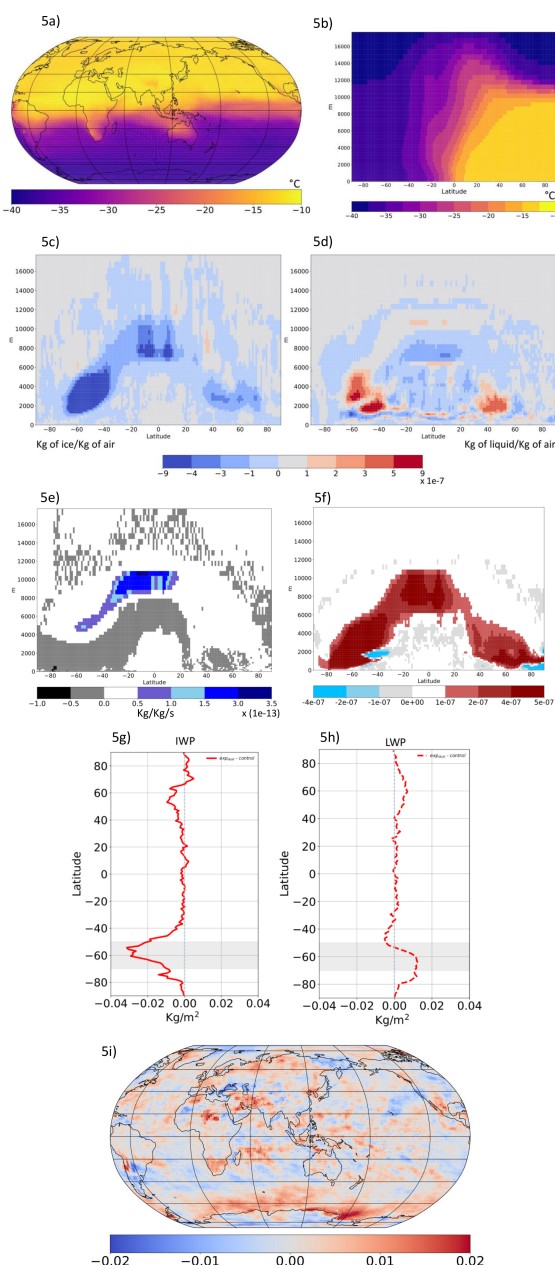

**Figure 5.** Presentation of results following the schematic. (5a) annual-mean spatial distribution of $thetr_n$ at the surface level in $exp_{dust}$. (5b) annual-mean zonal average of $thetr_n$ depicting the vertical distribution in $exp_{dust}$. (5c) annual-mean anomaly in the ice cloud condensate from convection scheme between $exp_{dust}$ and $control$. (5d) Similar to 5c but for liquid cloud condensate. (5e) annual-mean anomaly in the heterogeneous nucleation rate between $exp_{dust}$ and $control$. (5f) annual-mean anomaly in the SCL between $exp_{dust}$ and $control$. (5g) and (5h) represent the zonally averaged anomalies between $exp_{dust}$ and $control$ in the IWP and LWP respectively over the stratocumulus boundary layer type clouds in the model. This is similar to Figs. (1a) and (1b) shown earlier. (5i) annual-mean anomaly in the in-cloud albedo between $exp_{dust}$ and $control$. Values calculated from daily-mean model output covering one year for Figs. 5a to 5d ; 20 years for Figs. (5e, 5f and 5i). Values calculated from 12-hourly instantaneous model output covering 20 years for Figs. 5g and 5h.


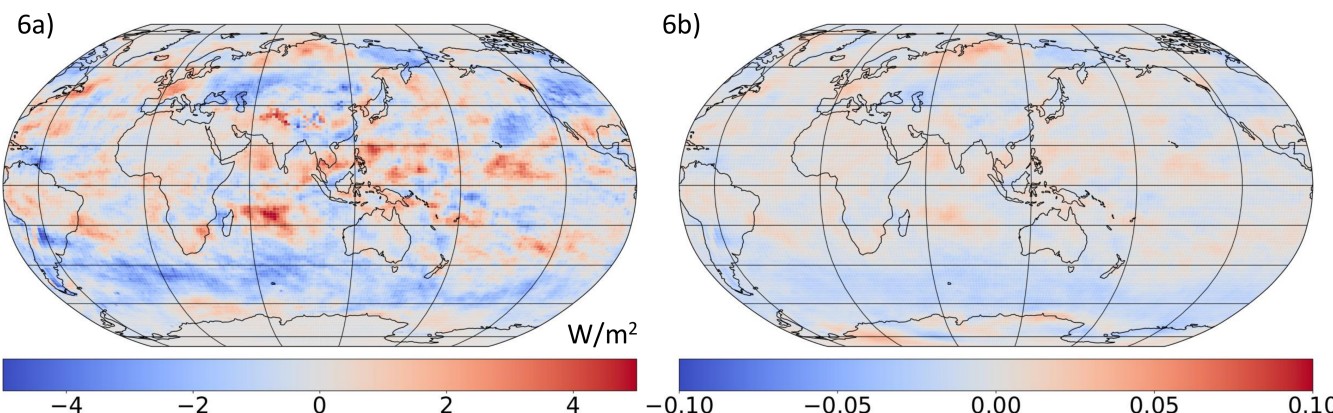

**Figure 6.** Annual-mean distributions of outgoing SW from TOA and total cloud cover (6a) Anomaly in the outgoing SW from TOA between $exp_{dust}$ and $control$ (6b) Anomaly in the total cloud cover between $exp_{dust}$ and $control$. Values calculated from daily-mean output over 20 years.



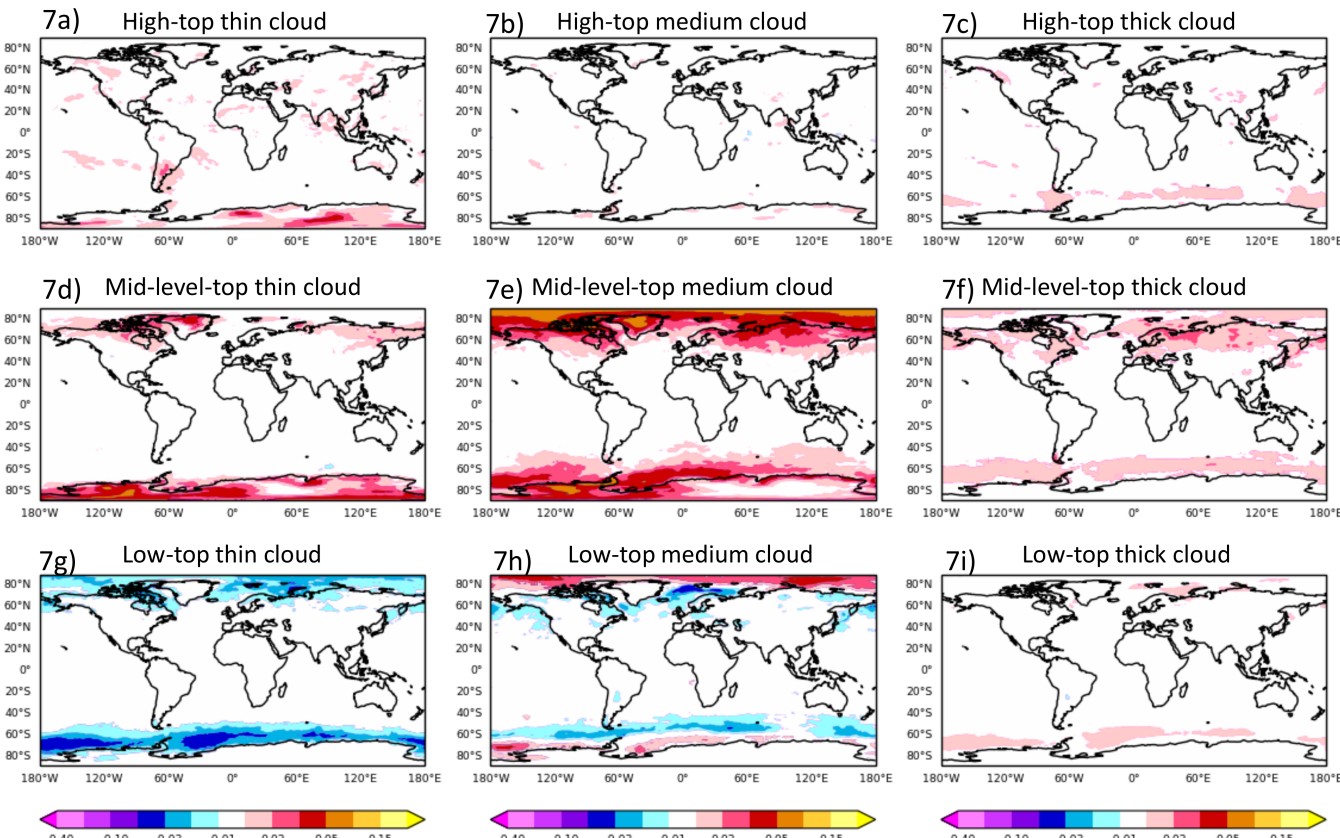

**Figure 7.** Anomaly in the annual-mean distributions of various cloud level types between $exp_{cap}$ and $control$. Plots are produced from similar experimental set-up mentioned in (Varma et al., 2020) but using a more technically recent version of the Unified Model with similar science configuration.

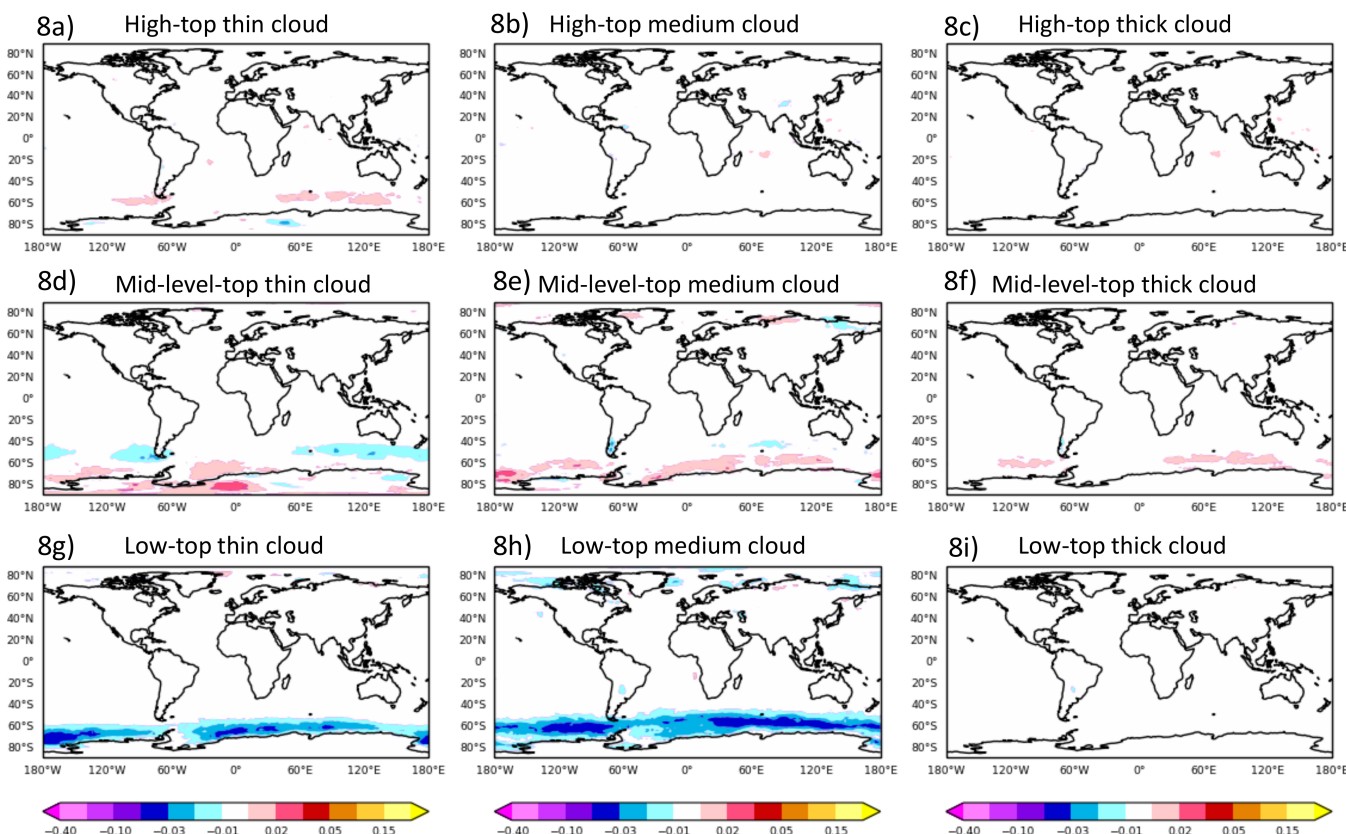

**Figure 8.** Similar to Fig. 7 but between $exp_{dust}$ and $control$

**Figure 9.** Similar to Fig. 7 but DJF anomaly between $exp_{eff}$ and $control_{eff}$ where the parameters in the convection scheme that could define the efficiency by which liquid and frozen cloud fractions for a given amount of detrained condensate is calculated (details in Section 5) are modified in both $exp_{dust}$ and $control$

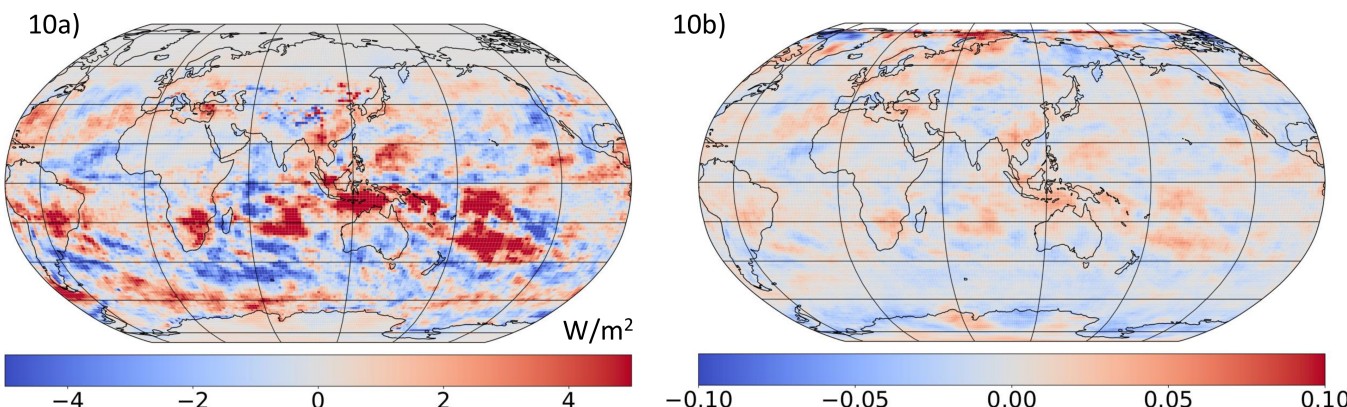

**Figure 10.** Similar to Fig. 6 but DJF anomaly between $exp_{eff}$ and $control_{eff}$ where the parameters in the convection scheme that could define the efficiency by which liquid and frozen cloud fractions for a given amount of detrained condensate is calculated (details in Section 5) are modified in both $exp_{dust}$ and $control$