# Peer review of "Introducing Ice Nucleating Particles functionality into the Unified Model and its impact on the Southern Ocean short-wave radiation biases"

_Atmospheric Chemistry and Physics, 2021_

## Referee Comment (RC1)

This manuscript performs global model simulations with a simplified dust-specific ice nucleation parameterization, which relates the activation temperature for immersion freezing to dust number concentrations, to investigate the low bias in outgoing shortwave radiation fluxes over the Southern Ocean (SO). After implementing the parameterization into the Met Office's Unified Model, more LWP and less IWP are simulated in the Southern Ocean (SO), along with an increase in cloud albedo. However, the outgoing shortwave radiation fluxes in SO are found to decrease, likely due to a reduction in cloud fraction, which makes the bias over the SO even worse. The authors conduct sensitivity experiments to investigate the cloud fraction decrease.

The question that the authors investigate is important and very interesting. However, unfortunately, the authors seem to have conceptual misunderstanding on the impact of aerosols on ice nucleation process, and thereby the dust-specific ice nucleation parameterization proposed and used in this manuscript is not valid. Besides, I have some concerns related to the interpretation of the results and experiments performed in the discussion. I therefore recommend rejection of this work.

General comments:

1. The statement that higher (lower) dust number density results in higher (lower) nucleation temperature is incorrect. It has been well established that the activation temperature for immersion freezing is related to aerosol species, instead of aerosol concentrations. As found by many observational studies, organic and biogenic aerosols tend to nucleate at warmer temperatures, while dust particles have lower activation temperatures. Therefore, the parameterization proposed in this paper that relates the activation temperature to dust concentrations is not valid. Even if the parameterization is valid, the authors should explain why they choose this formula and evaluate it against observations. This is the major reason for my rejection of this work. It is also not clear to me why the authors link the dust concentrations and the activation temperature of heterogeneous ice nucleation to the detrainment temperature in convection scheme. In other words, how is the detrainment process related to primary ice formation in the convection system?

Actually, there are many dust-specific ice nucleation parameterizations that are ready to use (e.g., Atkinson et al., 2013; DeMott et al., 2015; Hoose et al., 2010; Knopf & Alpert, 2013; Niemand et al., 2012; Ullrich et al., 2017; Wang et al., 2014). These parameterizations are derived based on either observational or theoretical evidences. They have also been implemented into regional and global models. The authors may want to use these parameterizations in their future work.

2. The authors should evaluate the modeling results against observations, before concluding if the new parameterization leads to any improvements in the model. For example, the authors can use MODIS LWP, CloudSat IWP, and MODIS cloud fraction. It would also be interesting to compare the simulated shortwave and longwave cloud forcing (SWCF and LWCF) with CERES-EBAF dataset. If possible, the authors may also evaluate the simulated dust and INPs in SO. For dust, the

authors can use CALIPSO dust extinction vertical profiles. For INPs, a lot of field measurements are available in SO, e.g., CAPRICORN campaign (McCluskey et al., 2018).

3. To investigate the cloud fraction decrease in $exp_{dust}$ over SO, the authors include the comparisons between $exp_{cap}$ and *control* in their discussion. However, $exp_{dust}$ and $exp_{cap}$ are two experiments with different modifications in the microphysical processes. What happened in $exp_{cap}$ should not be expected in $exp_{dust}$. Therefore, such comparisons do not help to understand the cloud fraction decrease in $exp_{dust}$. The authors should instead look into the changes in RH, precipitation, and probably lower-tropospheric stability (LTS) in $exp_{dust}$.

4. The sensitivity experiment, $exp_{eff}$, is not carefully designed. Why do you assume the liquid clouds are equally spread as the ice cloud in the convection scheme? Does this assumption make the model more physically correct? Are there any previous literatures that can support your assumption? Also, it is not fair to compare the DJF results in $exp_{eff}$ with the annual mean results in $exp_{dust}$.

Other comments:

Line 38: "… can proceed quicker …". It should be "proceed at warmer temperatures".

Section 2: It would be better to include how dust is parameterized in this section.

Section 3: The word "prognostic-dust parameterization" in the title of this section sounds like a dust transport parameterization. Please consider to replace it by something like "dust-specific ice nucleation parameterization".

Eq (1). How do you get the ice nucleation concentrations or the immersion freezing rate from $thetr_n$.

Line 162: "…, probably accounting for … than before". This sentence is not clear to me.

Line 164: Why do you show the IWP and LWP for stratocumulus boundary layer clouds only? Why not show those for the whole column?

Line 211-213: How do you know the liquid cloud fraction is smaller than the ice cloud fraction? What about the mixed-phase clouds? Also, the explanation in the second sentence does not make any sense.

Figure 6. It would be better to give a subtitle for each panel.

Reference

Atkinson, J. D., Murray, B. J., Woodhouse, M. T., Whale, T. F., Baustian, K. J., Carslaw, K. S., Dobbie, S., O'Sullivan, D., & Malkin, T. L. (2013). The importance of feldspar for ice nucleation by mineral dust in mixed-phase clouds. *Nature*, *498*(7454), 355–358. https://doi.org/10.1038/nature12278

DeMott, P. J., Prenni, A. J., McMeeking, G. R., Sullivan, R. C., Petters, M. D., Tobo, Y., Niemand, M., Möhler, O., Snider, J. R., Wang, Z., & Kreidenweis, S. M. (2015). Integrating laboratory and field data to quantify the immersion freezing ice nucleation activity of mineral dust particles. *Atmospheric Chemistry and Physics*, *15*(1), 393–409. https://doi.org/10.5194/acp-15-393-2015

Hoose, C., Kristjánsson, J. E., Chen, J.-P., & Hazra, A. (2010). A Classical-Theory-Based Parameterization of Heterogeneous Ice Nucleation by Mineral Dust, Soot, and Biological Particles in a Global Climate Model. *Journal of the Atmospheric Sciences*, *67*(8), 2483–2503. https://doi.org/10.1175/2010JAS3425.1

Knopf, D. A., & Alpert, P. A. (2013). A water activity based model of heterogeneous ice nucleation kinetics for freezing of water and aqueous solution droplets. *Faraday Discussions*, *165*, 513. https://doi.org/10.1039/c3fd00035d

McCluskey, Hill, T. C. J., Humphries, R. S., Rauker, A. M., Moreau, S., Strutton, P. G., Chambers, S. D., Williams, A. G., McRobert, I., Ward, J., Keywood, M. D., Harnwell, J., Ponsonby, W., Loh, Z. M., Krummel, P. B., Protat, A., Kreidenweis, S. M., & DeMott, P. J. (2018). Observations of Ice Nucleating Particles Over Southern Ocean Waters. *Geophysical Research Letters*, *45*(21), 11,989-11,997. https://doi.org/10.1029/2018GL079981

Niemand, M., Möhler, O., Vogel, B., Vogel, H., Hoose, C., Connolly, P., Klein, H., Bingemer, H., DeMott, P., Skrotzki, J., & Leisner, T. (2012). A Particle-Surface-Area-Based Parameterization of Immersion Freezing on Desert Dust Particles. *Journal of the Atmospheric Sciences*, *69*(10), 3077–3092. https://doi.org/10.1175/JAS-D-11-0249.1

Ullrich, R., Hoose, C., Möhler, O., Niemand, M., Wagner, R., Höhler, K., Hiranuma, N., Saathoff, H., & Leisner, T. (2017). A New Ice Nucleation Active Site Parameterization for Desert Dust and Soot. *Journal of the Atmospheric Sciences*, *74*(3), 699–717. https://doi.org/10.1175/JAS-D-16-0074.1

Wang, Y., Liu, X., Hoose, C., & Wang, B. (2014). Different contact angle distributions for heterogeneous ice nucleation in the Community Atmospheric Model version 5. *Atmospheric Chemistry and Physics*, *14*(19), 10411–10430. https://doi.org/10.5194/acp-14-10411-2014

---

## Author Comment (AC1)

**Reviewer 1 comments**

**Summary**

This manuscript performs global model simulations with a simplified dust-specific ice nucleation parameterization, which relates the activation temperature for immersion freezing to dust number concentrations, to investigate the low bias in outgoing shortwave radiation fluxes over the Southern Ocean (SO). After implementing the parameterization into the Met Office's Unified Model, more LWP and less IWP are simulated in the Southern Ocean (SO), along with an increase in cloud albedo. However, the outgoing shortwave radiation fluxes in SO are found to decrease, likely due to a reduction in cloud fraction, which makes the bias over the SO even worse. The authors conduct sensitivity experiments to investigate the cloud fraction decrease.

The question that the authors investigate is important and very interesting. However, unfortunately, the authors seem to have conceptual misunderstanding on the impact of aerosols on ice nucleation process, and thereby the dust-specific ice nucleation parameterization proposed and used in this manuscript is not valid. Besides, I have some concerns related to the interpretation of the results and experiments performed in the discussion. I therefore recommend rejection of this work.

**Specific Points**

The statement that higher (lower) dust number density results in higher (lower) nucleation temperature is incorrect. It has been well established that the activation temperature for immersion freezing is related to aerosol species, instead of aerosol concentrations. As found by many observational studies, organic and biogenic aerosols tend to nucleate at warmer temperatures, while dust particles have lower activation temperatures. Therefore, the parameterization proposed in this paper that relates the activation temperature to dust concentrations is not valid. Even if the parameterization is valid, the authors should explain why they choose this formula and evaluate it against observations. This is the major reason for my rejection of this work.

We would like to thank the reviewer for the very helpful review. After going through the comments, we realised that the message we were trying to convey has not been captured well in the submitted version of the manuscript. So, we have restructured the manuscript to give more clarity to the focus of this study. We hope that this will address many of the reviewer comments as well. We would like to clarify that we were not trying to include a new INP parametrisation as such in the Unified Model rather introducing a workaround for the lack of INP functionality in the global version of our model. This study is more like a follow up to the Varma et al., 2020 in the sense that Varma et al., 2020 had a hemispherical impact in terms of supercooled liquid content and cloud brightness. We wanted to make a targeted response over the SO region alone. As now mentioned and emphasized in the revised manuscript, if there already was an INP recognised in the model, this would have been relatively straightforward. Since the Unified Model (at the resolution we use) does not have an INP parametrisation presently available, we just implemented the prognostic dust approach so that there is more regional distribution of heterogeneous nucleation temperatures and hence ice/liquid cloud formation. As a result of this parametrisation, we now have more targeted response on SCL and in-cloud albedo over the SO region. We now have restructured the manuscript to take these into account. Most importantly, we have also added an interim comparison of our parametrisation with that of Demott et al., 2010 to show that they are agreeable.

It is also not clear to me why the authors link the dust concentrations and the activation temperature of heterogeneous ice nucleation to the detrainment temperature in convection scheme. In other words, how is the detrainment process related to primary ice formation in the convection system?

As noted in Section 3, since in-cloud micro-physics and convective clouds are treated separately in almost all of the low-resolution GCMs, immersion freezing is also parametrized separately in micro-physics and convection schemes. While in-cloud micro-physics predicts the cloud phase, the convection scheme provides a temperature dependent threshold for the detraining of ice (e.g. Kay et al 2016). As a result, along with the micro-physics scheme, the convection scheme also plays a role in determining the ice formation in the model through detrainment temperatures.

Actually, there are many dust-specific ice nucleation parameterizations that are ready to use (e.g., Atkinson et al., 2013; DeMott et al., 2015; Hoose et al., 2010; Knopf and Alpert, 2013; Niemand et al., 2012; Ullrich et al., 2017; Wang et al., 2014). These parameterizations are derived based on either observational or theoretical evidences. They have also been implemented into regional and global models. The authors may want to use these parameterizations in their future work.

As added now in the Introduction section, in order to implement and thoroughly examine the impact of dust as INP on cloud radiation properties as per these existing parametrisations, ideally state-of-the-art atmospheric models with extensive double-moment bulk micro-physics schemes or comprehensive aerosol models that allow the identification of aerosol species and number densities etc are desired. However, for low-resolution GCMs (like ours), this is not currently available. Our model does not identify the dust species or number densities but rather provide the mass mixing ratios based on representative diameters belonging to 6 size bins. This makes any direct comparison practically impossible. Also, we have clarified that our motive hence is not a new INP parametrisation. However, there are currently ongoing developments on the implementation of a GLOMAP dust scheme (which allows the speciation of dust and use/comparison of some of the existing dust INP parametrisations feasible in the future). Also, please see the response to 'Specific Points'.

The authors should evaluate the modeling results against observations, before concluding if the new parameterization leads to any improvements in the model. For example, the authors can use MODIS LWP, CloudSat IWP, and MODIS cloud fraction. It would also be interesting to compare the simulated shortwave and longwave cloud forcing (SWCF and LWCF) with CERES-EBAF dataset. If possible, the authors may also evaluate the simulated dust and INPs in SO. For dust, the authors can use CALIPSO dust extinction vertical profiles. For INPs, a lot of field measurements are available in SO, e.g., CAPRICORN campaign (McCluskey et al., 2018).

We have now added comparison of SW TOA/LWCF/SWCF with CERES data in the Supplementary material.

To investigate the cloud fraction decrease in $exp_{dust}$ over SO, the authors include the comparisons between expcap and control in their discussion. However, expdust and expcap are two experiments with different modifications in the microphysical processes. What happened in expcap should not be expected in expdust. Therefore, such comparisons do not help to understand the cloud fraction decrease in expdust. The authors should instead look into the changes in RH, precipitation, and probably lower-tropospheric stability (LTS) in expdust.

We have now removed the results from the capacitance experiment in the Discussion section. We have also moved the additional experiment to the Supplementary material.

The sensitivity experiment, expeff, is not carefully designed. Why do you assume the liquid clouds are equally spread as the ice cloud in the convection scheme? Does this assumption make the model more physically correct? Are there any previous literatures that can support your assumption? Also, it is not fair to compare the DJF results in expeff with the annual mean results in expdust.

We have now completely removed the mention of $exp_{eff}$. As the model tuning is an ongoing process, we have added another sensitivity study with results included in the Supplementary material.

**Other comments**

Line 38: "... can proceed quicker ...". It should be "proceed at warmer temperatures".

Modified

Section 2: It would be better to include how dust is parameterized in this section.

Section 3: The word "prognostic-dust parameterization" in the title of this section sounds like a dust transport parameterization. Please consider to replace it by something like "dust-specific ice nucleation parameterization".

We have changed the section heading to "Nucleation temperature as function of dust distribution: Experimental design"

Eq (1). How do you get the ice nucleation concentrations or the immersion freezing rate from thetrn.

As mentioned earlier, we have now made it clear that we are not introducing an INP parametrisation. Also, please see the additional section in the Supplementary material showing comparison with Demott et al., 2010.

Line 162: "..., probably accounting for ... than before". This sentence is not clear to me.

The sentence has been modified.

Line 164: Why do you show the IWP and LWP for stratocumulus boundary

layer clouds only? Why not show those for the whole column?

The focus of IWP/LWP in this study is on the stratocumulus boundary layer type clouds (in lines with Varma et al., 2020 study). We have now made it clear in the manuscript. The IWP/LWP plots through the entire cloud types/column are now included in the Supplementary material.

Line 211-213: How do you know the liquid cloud fraction is smaller than the ice cloud fraction? What about the mixed-phase clouds? Also, the explanation in the second sentence does not make any sense.

This has been modified.

Figure 6. It would be better to give a subtitle for each panel.

Added

**References**

Atkinson, J. D., Murray, B. J., Woodhouse, M. T., Whale, T. F., Baustian, K. J., Carslaw, K. S., Dobbie, S., O'Sullivan, D., Malkin, T. L. (2013). The importance of feldspar for ice nucleation by mineral dust in mixed-phase clouds. Nature, 498(7454), 355–358. https://doi.org/10.1038/nature12278

DeMott, P. J., Prenni, A. J., McMeeking, G. R., Sullivan, R. C., Petters, M. D., Tobo, Y., Niemand, M., Möhler, O., Snider, J. R., Wang, Z., Kreidenweis, S. M. (2015). Integrating laboratory and field data to quantify the immersion freezing ice nucleation activity of mineral dust particles. Atmospheric Chemistry and Physics, 15(1), 393–409. https://doi.org/10.5194/acp-15-393-2015

Hoose, C., Kristjánsson, J. E., Chen, J.-P., Hazra, A. (2010). A Classical-Theory-Based Parameterization of Heterogeneous Ice Nucleation by Mineral Dust, Soot, and Biological Particles in a Global Climate Model. Journal of the Atmospheric Sciences, 67(8), 2483– 2503. https://doi.org/10.1175/2010JAS3425.1

Knopf, D. A., Alpert, P. A. (2013). A water activity based model of heterogeneous ice nucleation kinetics for freezing of water and aqueous solution droplets. Faraday Discussions, 165, 513. https://doi.org/10.1039/c3fd00035d

McCluskey, Hill, T. C. J., Humphries, R. S., Rauker, A. M., Moreau, S., Strutton, P. G., Chambers, S. D., Williams, A. G., McRobert, I., Ward, J., Keywood, M. D., Harnwell, J., Ponsonby, W., Loh, Z. M., Krummel, P. B., Protat, A., Kreidenweis, S. M., DeMott, P. J. (2018). Observations of Ice Nucleating Particles Over Southern Ocean Waters. Geophysical Research Letters, 45(21), 11,989-11,997. https://doi.org/10.1029/2018GL079981

Niemand, M., Möhler, O., Vogel, B., Vogel, H., Hoose, C., Connolly, P., Klein, H., Bingemer, H., DeMott, P., Skrotzki, J., Leisner, T. (2012). A Particle-Surface-Area-Based Parameterization of Immersion Freezing on Desert Dust Particles. Journal of the Atmospheric Sciences, 69(10), 3077–3092. https://doi.org/10.1175/JAS-D-11-0249.1

Ullrich, R., Hoose, C., Möhler, O., Niemand, M., Wagner, R., Höhler, K., Hiranuma, N., Saathoff, H., Leisner, T. (2017). A New Ice Nucleation Active Site Parameterization for Desert Dust and Soot. Journal of the Atmospheric Sciences, 74(3), 699–717. https://doi.org/10.1175/JAS-D-16-0074.1

Wang, Y., Liu, X., Hoose, C., Wang, B. (2014). Different contact angle distributions for heterogeneous ice nucleation in the Community Atmospheric Model version 5. Atmospheric Chemistry and Physics, 14(19), 10411–10430. https://doi.org/10.5194/acp- 14-10411-2014

---

## Author Comment (AC2)

**Reviewer 2 comments**

**Summary**

In the manuscript titled "Introducing Ice Nucleating Particles functionality into the Unified Model and its impact on the Southern Ocean short-wave radiation biases", Varma et al. aim to implement an updated heterogeneous ice nucleation parameterization into the Unified Model in order to represent varying abundancies of ice nucleating particles. Ultimately, the goal of this study was the reduce shortwave radiation biases over the Southern Ocean, which is an active area of research.

Varma propose and implement a heterogenous ice nucleation parameterization approach that scales the primary ice nucleation temperature based on dust number concentrations. In comparing a 20-year simulation with the default physics (control) to 20-year simulation with the updated ice nucleation scheme, they report an overall decrease in ice water and increase in liquid water. Their results indicate simulated outgoing SW radiation decreased and cloud fraction also decreased with the proposed ice nucleation scheme. Varma et al. attribute these findings to additional feedbacks associated with convective cloud physics scheme, including a parameter that defines cloud fractions associated with a given amount of detrained condensation that differs between liquid and frozen clouds. The authors report that this approach 1) "improves the physics of the model" (line 234) and 2) "is one of the first global atmospheric models to implement such an approach to simulated the roles of INPs with minimum complexity in the micro-physics scheme to improve the SW radiation biases over the SO region".

While the topics of southern ocean ice nucleation, cloud phase, and SW radiation are very important, I have major concerns with this study. If my comments were to be adequately addressed, I think the study would change entirely and need to be resubmitted as a separate publication. Therefore, I recommend this manuscript to be rejected.

**General Comments**

Major Comment 1 – A major concern of this study is the approach for representing heterogeneous ice nucleation. The proposed parametrization essentially scales the heterogeneous ice nucleation temperature (thetr) based on dust number concentrations relative to an arbitrary reference dust concentration (refdust), such that thetr is lower (higher) for mixed phase clouds in regions with dust concentrations lower (higher) than refdust. refdust is completely arbitrary, referred to as a "tuning parameter", and is mentioned (line 121) to be a heuristic parametrization. A first major question is why was an existing dust ice nucleating particle parameterization not considered for this study? Neimand et al., 2012 and DeMott et al., 2015 are well-vetted dust-specific INP parameterizations that have been tested against laboratory and field measurements and have also been implemented into global models (Zhao et al., 2020). While the authors appear optimistic that observations could help constrain parameters in their thetr paramterization, it is not obvious how one would constrain non-physical parameters like thetr and refdust. This approach is likely highly sensitive to refdust, though the sensitivity of the study results to refdust is not tested here. Another major question is, given the number of challenges involved in simulating aerosol and ice nucleating particles, the parameterization needs to be assessed for accuracy and skill. There are observations of ice nucleating particles over the Southern Ocean that not included in this study, and I do not see a path in which this parameterization could be evaluated. It is also not clear how accurate simulated dust concentrations are in the Unified Model.

We would like to thank the reviewer for the very helpful review. We now added to the Introduction section that, in order to implement and thoroughly examine the impact of dust as INP on cloud radiation properties as per these existing parametrisations, ideally state-of-the-art atmospheric models with extensive double-moment bulk micro-physics schemes or comprehensive aerosol models that allow the identification of aerosol species and number densities etc are desired. However, for low-resolution GCMs (like ours), this is not currently available. Our model does not identify the dust species or number densities but rather provide the mass mixing ratios based on representative diameters belonging to 6 size bins. This makes any direct one-on-one comparison practically impossible. However, we have included an interim comparison with the Demott et al 2010 in the Supplementary material now. There are currently ongoing developments on the implementation of a GLOMAP dust scheme (which allows the speciation of dust and use/comparison of some of the existing dust INP parametrisations feasible in the future).

Major Comment 2 – I also have a major concern regarding the authors arguing that adding this new approach improves the model physics. First, I do not think the parameterization is in any way based on our understanding of the physical process of ice nucleation (i.e., for a given temperature, the available

number of ice nucleating particles depends on the aerosol population including the abundance and composition). I also do not see how the parameterization can be physically constrained. Regardless, I also do not see an assessment of the model performance with the "improved" physics. I only see analyses on changes in simulated clouds from the control simulation. Without an assessment of simulated clouds or TOA SW radiation, it is not clear that the model has actually been improved.

We have restructured and modified the paper to convey our message clearly in the recent version along with the added details (like comparison with observational data and model fluxes, Demott et al 2010 etc) in the Supplementary material. In terms of the elaborate assessment of the improved model physics, it can only come through improving the bias and rmse. But usually, in GCMs, several other aspects need to be tested and changed as well that ensure the climate model is radiatively balanced.

Major Comment 3 – I had a difficult time understanding what the results were from this study. While I understand that overall the liquid water path increased and ice water path decreased over the Southern Ocean region in the modified model compared to the control simulation, the overall bias was not assessed. I also do not understand the additional simulations used to assess changes in cloud cover and the convective scheme. There is an overall lack of explanation regarding why specific things were tested and what these could tell us. Additionally, there are many details missing that would limit the study from being reproducible, including how cloud level types are defined, what parameters in the convective scheme are changed and to what values.

We have now modified the manuscript content in general so that the focus of the study is highlighted clearly, which is to have a more targeted distribution of SCL over the SO region compared to our earlier capacitance study (Varma et al 2020). We have also restructured the Discussion section by moving some of the earlier content to the Supplementary material.

**Specific Comments**

Introduction:
    L44 – "Our focus will be on the immersion freezing process as it is the most commonly implemented heterogeneous ice nucleation process in global climate models (GCMs)." It is unclear to me what is meant by this. Do you mean that the immersion freezing process is most active in GCMs? Is it the case that immersion freezing is expected to be a common ice nucleation pathway in these low-level stratocumulus clouds? Do you have a reference for this?

Modified the statement.

L53 – "Among these, mineral dust is. . . " I agree that mineral dust is a strong source of ice nucleating particles near land sources, but many studies have highlighted the role of marine ice nucleating particle sin remote regions, especially the Southern Ocean (Burrows et al., 2013, Wilson et al., 2015, McCluskey et al., 2019). Why are marine sources ignored in this study?

Although we acknowledge the significance of PMOA as a potential INP, in the model version that we use,the marine organic emission is just lumped in terms of tracers and to make the link with the INP and the marine organic number concentration is not possible at this stage.

L54 – "in the generally low INP environment over the SO region" – how low? Please provide a reference (e.g., McCluskey et al., 2018; McFarquhar et al., 2020; Schmale. Et al., 2021)

We have modified the Introduction section.

L55 – ". . . INP dependency on immersion freezing is not included in most of the GCMs" – I believe the authors mean "immersion freezing dependency on INPs". Note that this is not necessarily the case anymore. See CESM2 ice nucleation scheme (e.g., Gettelman et al., 2019) which is based on the classical nucleation theory approach for estimating ice nucleation rates based on dust aerosol abundance from Hoose et al., 2010 and implemented into CESM by Wang et al., 2014. (note this CNT parametrization is referred to later, Line 65)

Modified the sentence

L67 – Just because mineral dust is a globally dominate in immersion freezing does not mean mineral dust dominates over the Southern Ocean region. Please include a discussion on the marine source of INPs.

Same as reply to comment on L53.

Methods:
    L87 – "paucity in INPs in clean environments. . . " – How low? Please include references.

Modified.

L92 – While the number density of dust is low over the SO region, studies have determined it possible for even small amounts of dust to have a strong influence on INP estimates (E.g., Zhao et al., 2020). How accurate is the simulated dust aerosol in regions far-removed from dust sources? Are there previous papers

that have asses the dust concentrations from this model, or possible observational datasets that could be used to make sure the dust concentrations are reasonable? Couldn't this can be particularly important over the remote ocean, where the INP number concentrations are extremely low and exceptionally sensitivity to transported aerosol?

As clarified in the modified version, we have adopted an approach as a workaround for the lack of INPs (dust or any other kind) in the current version of the model. Dust is only chosen so as to give the nucleation temperatures a hemispherical asymmetry for a targeted formation of SCL over the SO region. In addition, our paramterisation as an interim INP workaround is compared with the Demott et al 2010 study (added in the Supplementary material)

L116 - Please see Major Comment 1 regarding the proposed IN parametrization.

- I do not follow how the authors segregated identified the cloud level types used in Figures 7, 8, and 9.

Added details in the figure title that they follow ISCCP cloud level types. Also removed Figs 7 and 9.

- The authors mention an additional simulation previously published by Varma et al. (2020) and use these simulations as an additional comparison. The details of those simulations, or reasoning for included them in this paper are not clear to me.

In the modified version of the manuscript, we have made it clear that the intent of this study is to have a targeted distribution of SCL over the SO region compared to the earlier Varma et al 2020 study. We have also removed the comparison with $exp_{cap}$ from the Discussion section and now use that experiment only to show that SCL is more confined to SO region in the new approach.

Results
    L150 – it is extremely difficult to interpret figure 5 due to their small size and font size.

Modified the figure.

L172 – I think the discussion regarding the reduction in TOA outgoing SW radiation  cloud fraction needs to be expanded. The authors state "This is probably because, previously, the large amounts of ice clouds were introducing compensating errors, which the new scheme now removes" – what compensating errors? Can the authors expand on this?

*Modified the sentence.*

Discussion:

L183 – What is the reason for comparing to the expcap study? Did I miss that? Because this is not clear, I do not really follow most of this discussion.

*Modified the Discussion section. expcap study is now only used as a reference for the motivation of this approach.*

L184 -How were the model data segregated identified the cloud level types used in Figures 7, 8, and 9? Are these statistically significant differences between the exp simulations and control simulations?

*Modified figure caption that now includes they are identified similar to that of the ISCCP cloud level types. Also, removed Figs 7 and 9.*

L192 – this paragraph regarding "potential feedback processes from the convections scheme" is confusing without the support analysis? I think there would be room for more discussion on this.

*Added details in the Supplementary material under 'Additional experiment'.*

L203 – The authors introduce two more simulations, but it is not clear to me what is being changed. Perhaps adding the simulation details to the table would help. What is the difference between expeff and controleff? How would one constrain the parameters that control cloud fraction (a non-physical quanity) and detrained condensate (perhaps physical, but difficult to isolate). What values were used in the expeff and controleff experiments?

*We have removed the $exp_{eff}$ and added more details in the Supplementary material.*

Conclusions

L226 – "this approach provides a more realistic representation of nucleation temperatures..." – how do you know this is more realistic? One needs some comparisons against observations to claim this.

*We have modified this to 'more realistic representation of SCL content'. In the study by Bodas Salcedo et al 2016, they have shown using cyclone composite analysis and observational data that SO is dominated by SCL and we have already added this reference.*

**References**

Burrows, S. M., Hoose, C., Pöschl, U. Lawrence, M. G. Ice nuclei in marine air: biogenic particles or dust? Atmospheric Chemistry and Physics 13, 245–267 (2013).

DeMott, P. J. et al. Integrating laboratory and field data to quantify the immersion freezing ice nucleation activity of mineral dust particles. Atmospheric Chemistry and Physics 15, 393–409 (2015).

Gettelman, A. et al. High Climate Sensitivity in the Community Earth System Model Version 2 (CESM2). Geophysical Research Letters 46, 8329–8337 (2019).

Hoose, C., Kristjánsson, J. E., Chen, J.-P. Hazra, A. A Classical-Theory-Based Parameterization of Heterogeneous Ice Nucleation by Mineral Dust, Soot, and Biological Particles in a Global Climate Model. Journal of the Atmospheric Sciences 67, 2483–2503 (2010).

Wang, Y., Liu, X., Hoose, C. Wang, B. Different contact angle distributions for heterogeneous ice nucleation in the Community Atmospheric Model version 5. Atmospheric Chemistry and Physics 14, 10411–10430 (2014).

Niemand, M. et al. A Particle-Surface-Area-Based Parameterization of Immersion Freezing on Desert Dust Particles. Journal of the Atmospheric Sciences 69, 3077–3092 (2012).

McCluskey, C. S. et al. Observations of Ice Nucleating Particles Over Southern Ocean Waters. Geophysical Research Letters 45, 11,989-11,997 (2018).

McCluskey, C. S., DeMott, P. J., Ma, P. âL. Burrows, S. M. Numerical Representations of Marine IceâNucleating Particles in Remote Marine Environments Evaluated Against Observations. Geophys. Res. Lett. 46, 7838–7847 (2019).

McFarquhar, G. M. et al. Observations of Clouds, Aerosols, Precipitation, and Surface Radiation over the Southern Ocean: An Overview of CAPRICORN, MARCUS, MICRE, and SOCRATES. Bulletin of the American Meteorological Society 102, E894–E928 (2021).

Schmale, Julia, et al. "Overview of the Antarctic circumnavigation expedition: Study of preindustrial-like aerosols and their climate effects (ACE-SPACE)." Bulletin of the American Meteorological Society 100.11 (2019): 2260-2283.

Wilson, T. W. et al. A marine biogenic source of atmospheric ice-nucleating particles. Nature 525, 234–238 (2015).

Zhao, X., Liu, X., Burrows, S. Shi, Y. Effects of Marine Organic Aerosols as Sources of Immersion-Mode Ice Nucleating Particles on High Latitude Mixed-Phase Clouds. https://acp.copernicus.org/preprints/acp-2020-674/ (2020) doi:10.5194/acp-2020-674.